# Gender Differences in the Longitudinal Linkages between Fear of COVID-19 and Internet Game Addiction: A Moderated Multiple Mediation Model

**DOI:** 10.3390/bs14080675

**Published:** 2024-08-03

**Authors:** Qing Liu, Bin Gao, Yuedong Wu, Bo Ning, Yufei Xu, Fuyou Zhang

**Affiliations:** 1School of Education, Shanghai Normal University, Shanghai 200234, China; 1000497857@smail.shnu.edu.cn (Q.L.); 1000529932@smail.shnu.edu.cn (B.G.);; 2Lab for Educational Big Data and Policymaking, Shanghai Normal University, Shanghai 200234, China; 3School of Marxism, Shanghai Normal University, Shanghai 200234, China

**Keywords:** fear of COVID-19, loneliness, depression, gender differences, internet game addiction

## Abstract

Background: The COVID-19 outbreak has profoundly affected the psychological well-being of university students globally. Previous studies have found a positive longitudinal link between fear of COVID-19 (FoC-19) and internet addiction. However, there is a notable gap in the literature regarding COVID-19-specific predictors and underlying mechanisms of internet gaming addiction (IGA). Methods: Integrating the compensatory internet use theory and social role theory as frameworks, a three-wave longitudinal approach was used in this study to test the hypothesized model. Data spanning a duration of one year were gathered from undergraduate students in China. From 2021 to 2022, we conducted online self-report surveys in the midst of the COVID-19 pandemic to obtain participants’ levels of FoC-19, loneliness, depressive symptoms, and IGA. Results: FoC-19 showed a longitudinal positive relationship with IGA. The longitudinal link between FoC-19 and IGA was mediated by loneliness and depression. Gender moderated the direct effect of FoC-19 on IGA, with this effect being significant only among male students and not among their female counterparts. Conclusion: These findings advance our comprehension of the mechanisms and gender differences underlying the link between FoC-19 and IGA, and provide a novel perspective for interventions to reduce IGA.

## 1. Introduction

The global spread of the coronavirus (COVID-19) pandemic constituted a significant stressor that detrimentally affected the mental well-being of college students worldwide [1,2]. In particular, in order to cope with and escape from COVID-19-induced pressures and challenges, some university students became addicted to social media software and online games. Given the significant growth in the number of online game players over the last decade, for example, China has become the world’s largest online gaming market, with approximately 370 million online gamers by 2023 [3]. Online gaming is an important way for people to relax and find entertainment. Nevertheless, excessive engagement in online gaming can result in problematic behaviors, including internet game addiction (IGA) [4,5]. IGA denotes persistent and recurrent internet gaming that significantly disrupts an individual’s daily life [6]. Significantly, the COVID-19 pandemic has exacerbated the prevalence of IGA globally [7,8]. Therefore, understanding how factors within the COVID-19 context relate to IGA and how these factors may be targeted for intervention is crucial [9]. In recent years, COVID-19-related fear may be of special concern for young people, and many empirical studies have confirmed a significant positive link between the fear of COVID-19 (FoC-19) and addictive behaviors (e.g., internet addiction, smartphone addiction) [10,11,12,13].

However, several research gaps persist, necessitating further investigation. First, most of these studies utilized cross-sectional methodologies, lacking longitudinal exploration of the temporal relationships between variables, thereby constraining our comprehension of the enduring impacts of FoC-19 on addictive behaviors. Second, the obscure mechanisms that connect FoC-19 to problematic addictive behaviors remain insufficiently understood. Furthermore, prior researchers investigating the correlation between COVID-19 fear, adverse feelings (such as loneliness and depression), and addictive behaviors failed to consider gender differences [12,14], potentially compromising the specificity of interventions by neglecting group disparities. Third, the bulk of research examining the adverse effects of FoC-19 has been conducted in Western countries, with a scarcity of research in Eastern collectivist cultural backgrounds, such as China. Specifically, China implemented a rigorous “dynamic zero COVID-19 strategy” during the pandemic, significantly impacting people’s daily lives, mobility, and work, unlike the situations observed in other nations. Therefore, this study aims to explore the longitudinal associations, underlying mechanisms, and gender differences between FoC-19 and IGA in the Chinese culture.

### 1.1. Fear of COVID-19 and IGA

The compensatory internet use theory (CIUT) suggests that problematic internet use behaviors, such as IGA, represent a maladaptive coping mechanism utilized by individuals to regulate adverse feelings [15]. CIUT posits that individuals turn to the internet to compensate for deficiencies in their offline lives or to escape from negative emotions and stressors. This theory provides a framework to understand how and why individuals, particularly young people, might engage in excessive online activities as a way of coping with life’s challenges. In the context of the COVID-19 pandemic, the uncertainty introduced into people’s lives, studies, and work, including the risk of infection and mortality, has led to a surge in negative emotions such as fear, loneliness, and depression [16]. The pandemic has disrupted traditional support systems and coping mechanisms, pushing individuals towards online spaces for solace and distraction. For young college students, who are already at a developmental stage marked by emotional volatility and a search for identity, the lure of online gaming becomes even stronger. It offers an immediate, albeit temporary, escape from the harsh realities imposed by the pandemic. By turning to online gaming, these students are seeking to mitigate and address their negative emotional experiences. This behavior aligns with CIUT’s assertion that problematic internet use is a response to unmet needs and emotional distress in the offline world.

Although no prior published research has directly examined the link between FoC-19 and IGA, there is indirect empirical support for this connection. For instance, recent cross-sectional studies have revealed a significant link between FoC-19 and internet addiction [10,11]. Therefore, given that IGA is a subtype of internet addiction, it is highly plausible that FoC-19 may induce IGA. Similarly, FoC-19 showed a significant positive link with the severity of cyberchondria [17]. Moreover, FoC-19 was associated with an increased risk of social media addiction and smartphone addiction among Chinese college students [18,19]. A study conducted with Turkish adolescents suggested that FoC-19 was more likely to trigger problematic internet use [20]. Based on the aforementioned findings, FoC-19 may be positively associated with IGA.

### 1.2. Mediating Role of Loneliness

Loneliness is recognized increasingly as a pivotal public health concern [21], particularly in the aftermath of the COVID-19 pandemic, and this phenomenon holds noteworthy implications for university students who, post-pandemic outbreak, have emerged as a susceptible demographic vulnerable to enduring psychosocial consequences. Specifically, the global response to the COVID-19 crisis prompted numerous universities to transition to online instruction [22], compelling students to adapt to prolonged periods of home confinement due to lockdown measures. This substantial disruption to their daily routines, coupled with a significant reduction in face-to-face social interactions, has markedly escalated the risk of experiencing feelings of isolation [1,23]. According to stress and coping theory [24], it suggests that individuals facing challenging life situations, such as the outbreak of the COVID-19 pandemic, may experience anxiety, distress, and alienation. This, in turn, can lead to addictive behaviors, particularly among those lacking self-confidence or effective coping skills. For such individuals, engaging in addictive behaviors functioned as a way to evade coping to alleviate stress, thereby perpetuating a detrimental cycle. Empirical evidence has highlighted loneliness as a crucial risk factor in predicting IGA [25]. Furthermore, loneliness emerged as a pivotal mediating factor, bridging various risk elements, such as parental loneliness, to the emergence of IGA [26]. Recent investigations have also highlighted the considerable predictive role of FoC-19 in fostering feelings of loneliness [12]. Simultaneously, several studies supported the significant contribution of loneliness to the initiation of IGA [25,27].

### 1.3. Mediating Role of Depression

Depressive tendencies among young individuals have surged over the last decade. For example, an extensive meta-analysis indicated a consistent uptick in depressive indicators among Chinese university students, reaching a recent detection rate of 20.8% from 2010 to 2020 [28,29]. Substantial empirical evidence emphasized the close association between FoC-19 and various adverse emotions, including stress, anxiety, and depression [30]. Additionally, depression acted as a mediating variable, linking risk factors (e.g., rumination, life events, and parental phubbing) to the emergence of internet addiction and IGA [31,32,33]. Recent research further established FoC-19 as a significant predictor of depression [34,35]. Meanwhile, multiple studies affirmed that depression played a substantial role in contributing to IGA [36,37,38].

### 1.4. Multiple Mediating Effects of Loneliness and Depression

According to the diathesis-stress model, individuals with a vulnerability may exhibit more depressive symptoms when exposed to adverse environments [39]. That is, college students with an emotional vulnerability (e.g., high loneliness) may exhibit more depressive symptoms when exposed to stressful events including the sudden outbreak of the COVID-19 pandemic. It is notable that the incidence of depressive manifestations among university students amounted to 56.8%, surpassing that of the general populace (29.1%) [40]. An empirical study focused on young adults conducted in the US discovered that depression rose during COVID-19, with alterations in loneliness being the primary factor for the increase in depression [41]. Also, numerous studies demonstrated that loneliness positively predicted depression [42,43]. Furthermore, a recent investigation substantiated the mediating role of loneliness in the prognostic association between fears associated with COVID-19 and depressive symptoms [16]. As mentioned above, both theoretical perspectives and empirical results suggest that loneliness and depression are important mediating roles that link risk factors to IGA.

### 1.5. Gender Differences

In fact, prior studies have indicated significant gender differences in FoC-19 [44,45], loneliness [46], depression [28], and addictive behaviors (e.g., smartphone addiction and IGA) [47,48]. Specifically speaking, a meta-analysis indicated that the COVID-19 pandemic had a more adverse impact on females, suggesting that females perceived COVID-19 as posing a greater risk to personal health and public well-being compared to males [44]. Moreover, women exhibited greater levels of loneliness and depression during the epidemic compared to men [46,49]. In contrast, research findings indicate that males exhibit a higher susceptibility to developing IGA compared to females [48,50]. Furthermore, gender could moderate the association between COVID-19 fear and social adjustment outcomes, such as human flourishing [51], life satisfaction [52], anxiety [44], and preventive health behavior [53]. However, few studies have empirically investigated the moderating role of gender in the association between FoC-19, loneliness, depression, and IGA.

Based on social role theory, gender role stereotypes have been found to influence online gaming behavior, with males being more susceptible to playing online games [54]. This theory posits that societal norms and expectations shape behaviors and attitudes according to gender. These gender roles are culturally constructed and dictate the appropriate behaviors for males and females, often leading to distinct behavioral patterns between genders. According to this theory, males are more likely to engage in competitive and skill-based activities, which are prevalent in online gaming. This inclination is partly due to societal expectations that encourage males to exhibit dominance, competitiveness, and mastery in various domains, including virtual environments. Consequently, males might find online gaming more appealing as it aligns with these gendered expectations and offers a platform to express these traits. Accordingly, we were primarily focused on examining the possible moderating role of gender in the associations between personal vulnerabilities (i.e., FoC-19, loneliness, and depression) and IGA.

### 1.6. The Present Study

Overall, this study made some notable contributions to the literature on IGA. First, it explored the longitudinal link between FoC-19 and IGA, addressing a notable research gap. Second, it used a three-wave longitudinal methodology to elucidate the underlying mechanisms by which FoC-19 influences IGA. Third, it extended the conceptual frameworks of CIUT and social role theory by integrating FoC-19 as a significant risk factor to enhance our comprehension of online problem behavior. Specifically, the conceptual model (see Figure 1) examined the potential mediating roles of loneliness and depression in the longitudinal effects of FoC-19 on IGA, as well as gender differences in these effects. Based on the aforementioned findings, we propose the following hypotheses:

**Hypothesis** **1.**
*FoC-19 may be positively associated with subsequent IGA.*


**Hypothesis** **2.**
*Loneliness independently mediates the link between FoC-19 and IGA.*


**Hypothesis** **3.**
*Depression independently mediates the link between FoC-19 and IGA.*


**Hypothesis** **4.**
*Loneliness and depression sequentially mediate the link between FoC-19 and IGA.*


**Hypothesis** **5.**
*Gender moderates the effect of FoC-19 on IGA (H5a), the effect of loneliness on IGA (H5b), and the effect of depression on IGA (H5c).*


## 2. Methods

### 2.1. Participants and Procedure

The study employed a convenience sampling method to enlist Chinese university students as participants, spanning a duration of one year, beginning in September 2021 amid the COVID-19 pandemic. The investigation involved the distribution of an online questionnaire at three distinct time intervals. Initially, 387 undergraduate participants completed the survey at Time 1. Subjects who completed the questionnaire hastily (in less than 10 min), provided incomplete information, or lacked experience in online gaming were disqualified during the data cleansing process. This curation process yielded a valid cohort of 325 respondents who reported their levels of FoC-19. At Time 2, the same cohort received an online survey, with 281 participants providing responses, effectively tracking their encounters with loneliness and depression. Lastly, at Time 3, 295 eligible respondents (M_age_ = 19.13, SD = 0.81) disclosed their levels of IGA, revealing an average of 2.64 h (SD = 1.78) of online gaming duration per day. Among these subjects, the gender composition comprised 116 males and 179 females. Regarding residence, 172 participants were from urban areas, while 153 were from rural areas. Concerning socioeconomic status, 26 participants reported a family monthly income of 3000 RMB or below, 245 participants reported an income between 3000 and 5000 RMB, 49 participants reported an income between 5000 and 8000 RMB, and 5 participants reported an income above 8000 RMB.

Data for this study were gathered through a Chinese online survey agency amidst the COVID-19 epidemic. Due to the research design adopting a longitudinal approach, we focused on the university student population, considering the accessibility of participants, potential attrition, and research costs. This study involved tracking participants for a duration of one year, collecting data at three distinct time points. Prior to completing the questionnaire items, subjects provided electronic informed consent. Engagement in the online questionnaire was entirely voluntary, and respondents had the option to withdraw at any stage following its completion. Ethical clearance for this investigation was secured from the ethics board at Shanghai Normal University.

### 2.2. Measurements

All survey items were presented in the Mandarin Chinese version, and the reliability and validity of these scales have been validated in prior studies. FoC-19 and IGA items were measured on a 5-point Likert scale (1 = strongly disagree, 5 = strongly agree), with higher scores implying greater FoC-19 and IGA. Loneliness and depression items were measured on a 4-point Likert scale (1 = never, 4 = always), with higher scores signifying greater loneliness and depression.

#### 2.2.1. Fear of COVID-19

Subjects’ fear of COVID-19 was assessed using the Fear Scale for COVID-19, which was initially created by Ahorsu et al. and consists of 7 items [55]. This scale was validated among Chinese college students [56]. The scale demonstrated good internal consistency (Cronbach’s α = 0.87) during Time 1 in our investigation.

#### 2.2.2. Loneliness

Loneliness was evaluated using the Brief Loneliness Scale [57], which was initially created by Hays and Dimatteo [58]. It consists of 8 items. The scale demonstrated favorable internal consistency (Cronbach’s α = 0.81) during Time 2 in the current investigation.

#### 2.2.3. The Patient Health Questionnaire-9 (PHQ-9)

The assessment of participants’ depressive symptoms employed the PHQ-9, created by Kroenke and Spitzer [59]. This uni-dimensional questionnaire consists of nine items. It was validated among college students in China [60]. The scale demonstrated excellent internal consistency (Cronbach’s α = 0.94) during Time 2.

#### 2.2.4. Internet Game Addiction

Participants’ IGA was assessed utilizing the Internet Game Addiction Scale, a subset derived from the Internet Addiction Scale created by Chinese researchers [61]. This scale comprises eight items. During Time 3 of our study, this scale demonstrated robust internal consistency (Cronbach’s α = 0.91).

### 2.3. Data Analysis

Descriptive and correlation analyses were conducted using SPSS version 24.0, and we employed the PROCESS macro (Models 4, 6, and 89) developed by Hayes et al. to examine the research hypothesis model [62]. The skewness and kurtosis values of the data were also assessed and the results showed that the data conformed to a normal distribution. We conducted Little’s MCAR test (χ^2^ (46) = 55.88, *p* = 0.15), which confirmed that the data were missing completely at random [63], and the average percentage of missing data for T2 and T3 was very close to 10%. We thus adopted listwise deletion as the method for handling missing data [64]. The bootstrapping method with 5000 resamples was utilized to evaluate the statistical significance of these effects. A significant indirect effect was indicated if the 95% confidence interval (CI) did not encompass zero. Prior studies have suggested associations between family socioeconomic status (SES), Hukou (0 = urban, 1 = rural), and age with IGA [6]. Thus, these factors were treated as covariates.

### 2.4. Testing the Common Method Bias (CMB)

To examine the impact of CMB on our findings, we conducted Harman’s single-factor test. A common variance analysis was conducted on the four surveys using factor analysis. Following principal component analysis, six eigenvalues exceeding one were extracted. The initial factor elucidating the variance was 31.82%, falling below the 40% threshold recommended by the critical standard [65], suggesting that CMB did not significantly affect the research data obtained from the self-report questionnaire.

## 3. Results

### 3.1. Preliminary Analyses

Initially, the present study conducted a comparative analysis of IGA levels and daily average online gaming duration across university students of varying genders at both T1 and T3. As illustrated in Table 1, the independent samples *t*-test demonstrated that female students exhibited significantly lower levels of IGA and a reduced daily average online gaming duration compared to male students.

Next, to explore the interrelations among all variables, bivariate correlations were conducted for each variable. As outlined in Table 2, the findings indicate a significant link between the gender of college students and IGA. Furthermore, the primary variables in the proposed research model, including FoC-19, loneliness and depression, and IGA, exhibited significant correlations with each other.

### 3.2. Testing the Measurement Model

This research extensively evaluated the measurement model’s fit by employing well-established indices such as the chi-square test, comparative fit index (CFI), Tucker–Lewis index (TLI), root mean square error of approximation (RMSEA), and standardized root mean square residual (SRMR). These metrics are well-regarded in the field as indicators of model adequacy. The overall fit of the measurement model was assessed using Mplus 8.3 software. Table 3 demonstrates that the goodness-of-fit indices for all scales indicated acceptable construct validity. These results collectively confirm the satisfactory fit of the measurement model across different scales in our study.

### 3.3. Testing the Mediation Model

We employed the PROCESS macro (Model 4) to explore the mediating influence of loneliness in the link between FoC-19 and IGA. The results (see Figure 2) indicated a significant positive impact of FoC-19 on loneliness (β = 0.28, *p* < 0.001), and loneliness exhibited a significant positive effect on IGA (β = 0.24, *p* < 0.001). Furthermore, the direct effect of FoC-19 on IGA was also significant (β = 0.13, *p* < 0.05). These findings showed that loneliness partially mediated the longitudinal link between FoC-19 and IGA (indirect effect = 0.07, 95%CI = [0.02, 0.12], contributing to 35% of the total effect (effect value = 0.20, 95%CI = [0.09, 0.31].

Likewise, utilizing the PROCESS macro (Model 4), we assessed the mediating effect of depression in the link between FoC-19 and IGA. The results (see Figure 3) demonstrated a significant positive link between FoC-19 and depression (β = 0.31, *p* < 0.001), and depression displayed a significant positive impact on IGA (β = 0.33, *p* < 0.001). However, the direct effect of FoC-19 on IGA was non-significant (β = 0.09, *p* > 0.05). These findings proved that depression functioned as a mediator in the link between FoC-19 and IGA (indirect effect = 0.10, 95%CI = [0.05, 0.17]), explaining 50% of the total effect (effect value = 0.20, 95%CI = [0.09, 0.31]).

Finally, we utilized the PROCESS macro (Model 6) to examine the sequential mediating effects of loneliness and depression in the link between FoC-19 and IGA. The results (see Figure 4) revealed that FoC-19 had a significant positive effect on loneliness (β = 0.29, *p* < 0.001) and depression (β = 0.19, *p* < 0.001). Similarly, loneliness had a significant positive effect on depression (β = 0.37, *p* < 0.001) and IGA (β = 0.13, *p* < 0.05). Besides, depression also had a significant positive effect on IGA (β = 0.29, *p* <0.001). However, the direct effect of FoC-19 on IGA was non-significant (β = 0.05, *p* > 0.05). These findings implied that loneliness and depression sequentially mediated in the longitudinal link between FoC-19 and IGA (indirect effect = 0.03, 95%CI = [0.01, 0.05]), accounting for 16.67% of the total effect (effect value = 0.18, 95%CI = [0.07, 0.29]).

### 3.4. Testing the Moderated Multiple Mediation Model

We utilized the PROCESS macro (Model 89) to investigate the moderated multiple mediation model. The statistical outcomes are delineated in Figure 5 and Table 4. Specifically, FoC-19 demonstrated significant predictive power for loneliness (β = 0.30, *p* < 0.001), depression (β = 0.19, *p* < 0.001), and IGA (β = 0.19, *p* < 0.05). Loneliness also exhibited a positive prediction for depression (β = 0.37, *p* < 0.001) and IGA (β = 0.13, *p* < 0.05). Additionally, the interaction term between FoC-19 and gender significantly predicted IGA (β = −0.27, *p* < 0.05). However, the product of loneliness and gender did not significantly predict IGA (β = 0.06, *p* > 0.05). Similarly, the interaction term between depression and gender did not significantly predict IGA (β = 0.14, *p* > 0.05). Hence, gender moderated the direct effect of FoC-19 on IGA. The results of all study hypothesis tests are shown in Table 5.

Following this, we performed straightforward slope tests to visually depict the moderating impact of gender. As depicted in Figure 6, the interaction plot revealed that among male students, there was a positive predictive association between FoC-19 and IGA (β = 0.19, *p* < 0.001), while among female students, this predictive relationship between FoC-19 and IGA was not significant (β = −0.07, *p* > 0.05).

## 4. Discussion

This research was the first attempt to extensively explore the longitudinal links and underlying mechanisms between FoC-19 and IGA. Through this empirical study, our findings have demonstrated that interventions to address the growing problem of IGA in university students need to concentrate on highlighting COVID-19 fear-induced adverse experiences and gender differences, indicating that male students are at an elevated risk for IGA. The current findings validate most of the research hypotheses. To contextualize our results within the broader research landscape, it is important to note that the psychological impacts of the COVID-19 pandemic have been widely documented, with numerous studies highlighting increased levels of anxiety, depression, and stress among various populations. Our study adds to this body of literature by specifically examining the role of FoC-19 in the development of IGA among university students. This aligns with previous research suggesting that the pandemic has exacerbated pre-existing mental health issues and introduced new stressors that contribute to maladaptive coping mechanisms, such as excessive online gaming. In the subsequent sections, we provide an extensive and insightful discussion of the important findings.

Correlation analysis confirmed positive longitudinal linkages between FoC-19 and IGA, supporting research Hypothesis 1, further expanding research findings in the field of addiction [10,11,12,13]. This result can be explained in several ways. First, lockdowns, social distancing, and other COVID-19-related restrictions may leave individuals with more free time [66], and online games offer a way to fill this time, providing entertainment and a sense of accomplishment as players progress through the game. Second, online games often have a social component, allowing individuals to connect with friends or make new social connections within the gaming community [67]. With social distancing policies in place during COVID-19, online games may function as a substitute for in-person social interactions. Third, online games may function as a coping mechanism for individuals experiencing heightened anxiety or fear related to COVID-19 [68]. Specifically, playing online games may be a way for people to cope with stress, temporarily alleviate negative emotions, and compensate for unmet social needs [69]. In other words, online games provide a virtual environment where individuals can immerse themselves in a different reality, temporarily distancing themselves from the fear and uncertainty associated with the pandemic.

Mediation analysis results showed that loneliness and depression acted as mediators between FoC-19 and IGA, which supported research Hypotheses 2–4. On the one hand, FoC-19 is a positive predictor of loneliness and depression, which is in accordance with the earlier findings [12,16]. Due to the COVID-19 pandemic, it was common for institutions of higher education to replace regular face-to-face instruction with online instruction [70]. However, this shift has created a host of mental health and academic issues for college students, like the absence of socialization, rising stress levels, increased anxiety, absence of social support, reduced employment opportunities, and increased employment pressure [71,72]. Furthermore, our findings revealed that loneliness significantly predicted depression, which is consistent with numerous previous studies [42,73]. On the other hand, loneliness and depression are both significant predictors of IGA. According to the self-medication hypothesis of addiction [74], addictive behaviors are considered a form of non-adaptive self-regulation when an individual is coping with a negative emotional or stressful state. Therefore, we can infer that university students who suffer from loneliness and depression as a consequence of COVID-19-related stressful events may use playing online games as an important way to deal with COVID-19-induced fears and regulate negative emotions.

The moderating effects analysis identified that gender moderates the link between FoC-19 and IGA, which supported research Hypothesis 5a. Specifically, male university students are more inclined to develop IGA when confronted with COVID-19-related fears. The following theoretical frameworks and empirical evidence may provide explanations for this outcome. First of all, the findings indicated that female undergraduate students not only reported having limited leisure time compared to men, but also experienced their free time in shorter intervals, engaging in digital gaming for briefer durations than their male counterparts [75]. Thus, male college students may dedicate more time to online gaming, and they may have an increased risk of suffering from IGA, especially in times of heightened fear and insecurity such as the COVID-19 pandemic. Drawing upon social role theory, researchers have discerned that gender role stereotypes can exert an influence on online gameplay [54]. Additionally, although the number of female gamers has been increasing in recent years, women still face many barriers when entering the gaming community due to gender stereotypes [76]. For example, the current portrayal of women in video game content remains biased, with female players continuing to contend with stereotypes, such as lack of competence in gameplay [77]. Moreover, a qualitative study indicated that female gamers in online gaming continue to encounter challenges stemming from stigmatized internal gender self-image [78]. Moreover, they may be inclined to disassociate themselves from the identity of being gamers, abstain from engaging in social gaming interactions, or refrain from participating in conventional online gaming activities.

### 4.1. Implications

Regarding the theoretical implications, this paper adopted the conceptual framework that integrates the CIUT proposed by Kardelfelt–Winther [15], and the social role theory articulated by Eagly [79]. Notably, previous studies have not simultaneously incorporated COVID-19-related predictors for IGA, such as COVID-19 fear, loneliness, and depression, within the CIUT and social role theory. As such, this study represents an extension of the CIUT and social role theory to a certain degree. Furthermore, the present study augments the social role theory by underscoring that male college students face a heightened susceptibility to IGA. Specifically, we elucidate the mediating processes and gender differences that connect COVID-19 fear and IGA by integrating variables such as loneliness and depression.

Practically speaking, school administrators should pay attention to the potential long-term risks posed by the COVID-19 pandemic. Relevant intervention measures should be dedicated to addressing the lasting negative impacts of the pandemic on the psychological well-being of college students, thereby ensuring a foundation for promoting both their mental health and academic development. Meanwhile, our study underscores the importance of considering specific characteristics of university students, such as gender, in the implementation of more targeted intervention measures for addressing IGA. Our findings suggest that interventions for IGA should focus on male university students, thereby enhancing the effectiveness and specificity of interventions. Furthermore, offering training and support, such as mindfulness training and self-compassion exercises, to enhance social support and positive coping skills can be useful for college students in managing negative emotions and reducing the likelihood of IGA. In addition to these recommendations for school administrators, policymakers and governments should consider implementing broader strategies to support the mental health of university students. This could include funding for mental health programs, developing policies that promote mental health awareness and education, and ensuring that universities have the resources necessary to provide comprehensive mental health services and academic support [80,81]. Moreover, policies that foster a supportive online environment and regulate the gaming industry to prevent addictive game designs could help mitigate the risks of IGA. By addressing both the individual and systemic factors contributing to IGA, these measures can create a more supportive environment for the mental health and academic success of college students.

### 4.2. Limitations and Future Research Directions

While this study contributes to our understanding of the longitudinal linkages between FoC-19 and IGA, several limitations should be acknowledged. Firstly, the sample primarily consisted of Chinese undergraduate students representing a collectivist culture, which limits the generalizability of the findings to other cultural contexts, such as individualistic cultures in Western developed countries. Future research should aim to replicate these findings in diverse cultures or countries to enhance the robustness of the results. Secondly, the use of self-report measures may introduce CMB and social desirability bias, potentially influencing the accuracy of the reported associations. Employing multi-method approaches, such as observational or experimental designs, could help mitigate these biases and provide more reliable data. Thirdly, while we focused on the direct effects and underlying mechanisms of FoC-19 on IGA, we recognize the potential for bidirectional relationships among these variables. Future studies could use a cross-lagged mediation model to explore these reciprocal associations. Furthermore, although gender differences were identified in the direct effect of FoC-19 on IGA, the reasons behind these differences remain unclear. Future investigation is warranted to explore the underlying pathways that contribute to gender disparities in the link between FoC-19 and IGA. Finally, our findings may inform the design of intervention strategies to mitigate the adverse effects of FoC-19 on young adults’ IGA, particularly by addressing FoC-19, loneliness, and depression, with a special focus on male students.

## 5. Conclusions

This study is the first attempt to utilize a longitudinal design to demonstrate a positive longitudinal association and the underlying mechanisms between FoC-19 and IGA. This study further expands the application of the CIUT and social role theory to the field of addictive behavior. Furthermore, loneliness and depression acted as mediating roles in the association between FoC-19 and IGA. Meanwhile, the moderating effect of gender suggests that the predictive effect of FoC-19 on IGA holds only for male students. These findings may inspire the design of intervention strategies to cope with the adverse effects of FoC-19 on young adults’ IGA.

## Figures and Tables

**Figure 1 behavsci-14-00675-f001:**
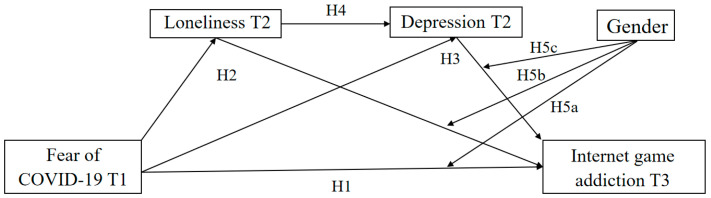
The conceptual model. Note. T1–T3 = waves of assessment from Time 1 to Time 3.

**Figure 2 behavsci-14-00675-f002:**
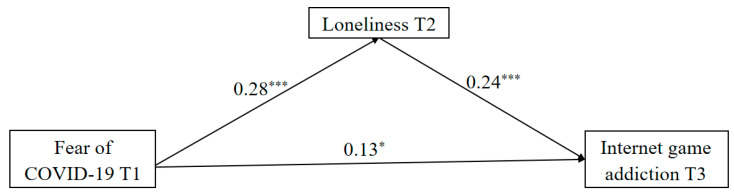
The mediating role of loneliness. Note: ** p* < 0.05, **** p* < 0.001.

**Figure 3 behavsci-14-00675-f003:**
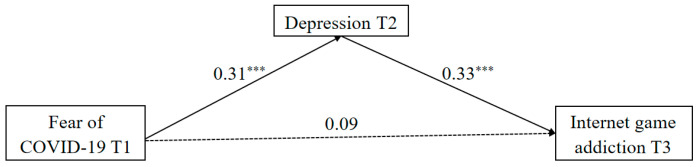
The mediating role of depression. Note: **** p* < 0.001. Dashed lines indicate insignificant path coefficients.

**Figure 4 behavsci-14-00675-f004:**
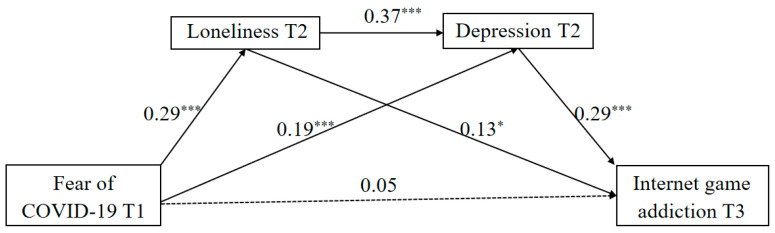
The multiple mediation model. Note: ** p* < 0.05,**** p* < 0.001. Dashed lines indicate insignificant path coefficients.

**Figure 5 behavsci-14-00675-f005:**
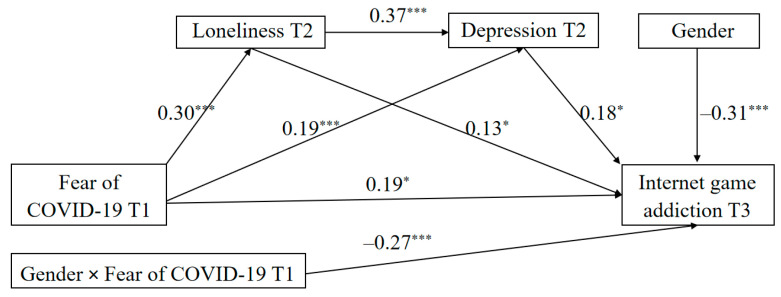
The moderated multiple mediation model. Note: The path coefficients in the figure are standardized. For brevity, the nonsignificant paths are not displayed. * *p* < 0.05, *** *p* < 0.001.

**Figure 6 behavsci-14-00675-f006:**
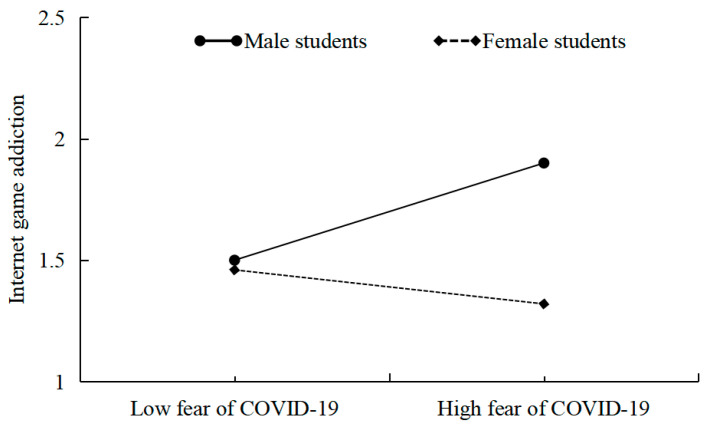
The interaction of gender and FoC-19 on IGA.

**Table 1 behavsci-14-00675-t001:** Comparative analysis of IGA levels and daily gaming duration by gender.

Variables	Gender	Time 1 (M ± SD)	*t* Value	Time 3 (M ± SD)	*t* Value
Hours/day of game play	Males	2.45 ± 1.75	6.33 ***	3.03 ± 2.12	3.51 **
	Females	1.33 ± 1.43		2.21 ± 1.64	
IGA	Males	2.39 ± 0.89	5.44 ***	2.39 ± 0.85	2.88 **
	Females	1.87 ± 0.80		2.11 ± 0.74	

Note: *** p* < 0.05, **** p* < 0.01, IGA = internet game addiction.

**Table 2 behavsci-14-00675-t002:** Descriptive statistics and correlation matrix.

Variables	M	SD	1	2	3	4	5
1. Gender	—	—	1				
2. FoC-19 (T1)	2.09	0.87	0.03	1			
3. Loneliness (T2)	2.00	0.53	0.05	0.29 **	1		
4. Depression (T2)	1.53	0.62	−0.11	0.28 **	0.42 **	1	
5. IGA (T3)	2.22	0.80	−0.17 **	0.20 **	0.28 **	0.38 **	1

Note: *** p* < 0.01. IGA = internet game addiction.

**Table 3 behavsci-14-00675-t003:** The goodness of fit of the measurement model.

	χ^2^/df	CFI	TLI	SRMR	RMSEA
1. FoC-19 scale	3.04	0.97	0.94	0.032	0.079
2. Loneliness scale	2.24	0.96	0.94	0.041	0.063
3. PHQ-9	2.92	0.95	0.93	0.027	0.078
4. IGA scale	2.75	0.95	0.94	0.036	0.074

**Table 4 behavsci-14-00675-t004:** Moderated multiple mediation model analysis results.

Predictors	Model 1 (Loneliness)			Model 2 (Depression)			Model 3 (IGA)		
	β	*t*	95% CI	β	*t*	95% CI	β	*t*	95% CI
SES	0.11	0.92	[−0.13, 0.34]	−0.06	−0.57	[−0.28, 0.16]	−0.22 *	−2.02	[−0.44, −0.01]
Age	0.12	2.23	[0.01, 0.23]	−0.07	−1.47	[−0.17, 0.03]	−0.01	−0.31	[−0.12, 0.08]
Hukou	−0.17	−1.40	[−0.41, 0.07]	−0.03	−0.29	[−0.26, 0.19]	−0.26	−2.31	[−0.49, −0.04]
FoC-19 T1	0.30 ***	5.12	[0.18, 0.41]	0.19 ***	3.40	[0.08, 0.31]	0.19 *	2.46	[0.04, 0.35]
Loneliness T2				0.37 ***	6.21	[0.25, 0.49]	0.13 *	2.30	[0.01, 0.25]
Depression T2							0.18 *	2.19	[0.02, 0.35]
Gender							−0.31 **	−2.66	[−0.52, −0.08]
FC-19 × Gender							−0.27 *	−2.42	[−0.49, −0.05]
*R^2^*	0.12			0.23			0.23		
*F*	8.64 ***			14.58 ***			8.13 ***		

Note: * *p* < 0.05, ** *p* < 0.01, *** *p* < 0.001. For brevity, insignificant path coefficients of interaction term are not presented in the figure.

**Table 5 behavsci-14-00675-t005:** Results of research hypothesis testing.

Hypotheses	Research Hypothesis Statement	Results
H1	FoC-19 may be positively associated with subsequent IGA.	Supported
H2	Loneliness independently mediates the link between FoC-19 and IGA.	Supported
H3	Depression independently mediates the link between FoC-19 and IGA.	Supported
H4	Loneliness and depression sequentially mediate the link between FoC-19 and IGA.	Supported
H5a	Gender moderates the effect of FoC-19 on IGA.	Supported
H5b	Gender moderates the effect of loneliness on IGA.	Not supported
H5c	Gender moderates the effect of depression on IGA.	Not supported

## Data Availability

The data of this study are available from the corresponding author upon reasonable request.

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
