# Peer review of "Gender Differences in the Longitudinal Linkages between Fear of COVID-19 and Internet Game Addiction: A Moderated Multiple Mediation Model"

_behavsci, 2024, doi:10.3390/bs14080675_

Round 1

Reviewer 1 Report

Comments and Suggestions for Authors

The conceptual framework should be expanded to include a deeper discussion of Compensatory Internet Use Theory (CIUT) and Social Role Theory. This will strengthen the theoretical basis of the study and allow for better contextualization of the findings. In addition, there is a need to review and expand the existing literature on fear of COVID-19 and Internet addictions, ensuring a more complete integration with previous studies and relevant theories. The hypotheses presented should be clearly derived from the literature review and theories discussed, ensuring greater clarity in their formulation and presentation.

It is essential to provide a more detailed description of the sample used in the study, including aspects of diversity and additional demographic characteristics beyond gender and age, such as socioeconomic status and educational background. A more robust justification for the use of convenience sampling should also be included, discussing the possible limitations this may imply for the generalizability of the results. In addition, the discussion of the validity and reliability of the scales used should be expanded, making reference to previous studies that have validated these tools in similar populations.

The results should include additional details on the statistical methods used to analyze the data, justifying the choice of the PROCESS macro and the specific models applied. It is necessary to provide a more detailed interpretation of the correlations found between the main variables, explaining how these relationships support or refute the hypotheses put forward. Data visualization can be improved by including additional graphs illustrating the key relationships and the mediation and moderation effects discussed in the study.

In the conclusions section, it is important to expand on the practical and theoretical implications, providing specific recommendations for future interventions based on the study findings. Also expand on the discussion of the limitations of the study, detailing how the cultural context and sampling may have influenced the results. Proposing clear directions for future research, including suggestions for additional longitudinal studies and alternative methodological approaches, will contribute to a better understanding of the phenomenon under investigation.

Comments on the Quality of English Language

It is crucial to ensure that the text is coherent and clear throughout, avoiding excessive use of jargon and explaining technical terms when necessary. Finally, it is recommended to verify that all citations and references are complete and correctly formatted according to the journal's style, thus ensuring the accuracy and professionalism of the article.

Author Response

Thank you very much for taking the time to review this manuscript. Please find the detailed responses below and the corresponding revisions in the re-submitted files.

Comments 1: The conceptual framework should be expanded to include a deeper discussion of Compensatory Internet Use Theory (CIUT) and Social Role Theory. This will strengthen the theoretical basis of the study and allow for better contextualization of the findings. In addition, there is a need to review and expand the existing literature on fear of COVID-19 and Internet addictions, ensuring a more complete integration with previous studies and relevant theories. The hypotheses presented should be clearly derived from the literature review and theories discussed, ensuring greater clarity in their formulation and presentation.

Response:We have further expanded the discussion of the Compensatory Internet Use Theory (CIUT) and Social Role Theory in the manuscript. Additionally, we enriched the existing literature on fear of COVID-19 and Internet addictions. Our research hypotheses are grounded in the literature review and theories presented in each section, ensuring a rigorous and credible foundation. All hypotheses are now clearly stated after the figure, ensuring a more logical flow and better organization of the content.

The compensatory internet use theory (CIUT) suggests that problematic internet use behaviors, such as IGA, represent a maladaptive coping mechanism utilized by individuals to regulate adverse feelings [15]. CIUT posits that individuals turn to the internet to compensate for deficiencies in their offline lives or to escape from negative emotions and stressors. This theory provides a framework to understand how and why individuals, particularly young people, might engage in excessive online activities as a way of coping with life’s challenges. In the context of the COVID-19 pandemic, the uncertainty introduced into people's lives, studies, and work, including the risk of infection and mortality, has led to a surge in negative emotions such as fear, loneliness, and depression [16]. The pandemic has disrupted traditional support systems and coping mechanisms, pushing individuals towards online spaces for solace and distraction. For young college students, who are already at a developmental stage marked by emotional volatility and a search for identity, the lure of online gaming becomes even stronger. It offers an immediate, albeit temporary, escape from the harsh realities imposed by the pandemic. By turning to online gaming, these students are seeking to mitigate and address their negative emotional experiences. This behavior aligns with CIUT’s assertion that problematic internet use is a response to unmet needs and emotional distress in the offline world.

Based on social role theory, gender role stereotypes have been found to influence online gaming behavior, with males being more susceptible to playing online games [55]. Social role theory posits that societal norms and expectations shape behaviors and attitudes according to gender. These gender roles are culturally constructed and dictate the appropriate behaviors for males and females, often leading to distinct behavioral patterns between genders. Social role theory suggests that males are more likely to engage in competitive and skill-based activities, which are prevalent in online gaming. This inclination is partly due to societal expectations that encourage males to exhibit dominance, competitiveness, and mastery in various domains, including virtual environments. Consequently, males might find online gaming more appealing as it aligns with these gendered expectations and offers a platform to express these traits.

Based on the aforementioned findings, we propose the following hypotheses:

Hypothesis 1: FoC-19 may be positively associated with subsequent IGA.

Hypothesis 2: Loneliness independently mediates the link between FoC-19 and IGA.

Hypothesis 3: Depression independently mediates the link between FoC-19 and IGA.

Hypothesis 4: Loneliness and depression sequentially mediate the link between FoC-19 and IGA.

Hypothesis 5: Gender moderates the effect of FoC-19 on IGA (H5a), the effect of loneliness on IGA (H5b), and the effect of depression on IGA (H5c).

Comments 2:  It is essential to provide a more detailed description of the sample used in the study, including aspects of diversity and additional demographic characteristics beyond gender and age, such as socioeconomic status and educational background. A more robust justification for the use of convenience sampling should also be included, discussing the possible limitations this may imply for the generalizability of the results. In addition, the discussion of the validity and reliability of the scales used should be expanded, making reference to previous studies that have validated these tools in similar populations.

Response:

Thank you for your valuable suggestions. We have expanded the description of the participants by including their residence and family socioeconomic status. Additionally, we have elaborated on the limitations of convenience sampling for the generalizability of the results in the discussion section. Furthermore, we have referenced relevant literature to verify the validity and reliability of the scales used in this study and included the results of the measurement model analysis along with the CFA fit indices for each scale. Regarding residence, 172 participants were from urban areas, while 153 were from rural areas. Concerning socioeconomic status, 26 participants reported a family monthly income of 3000 RMB or below, 245 participants reported an income between 3000 and 5000 RMB, 49 participants reported an income between 5000 and 8000 RMB, and 5 participants reported an income above 8000 RMB.

Comments 3: The results should include additional details on the statistical methods used to analyze the data, justifying the choice of the PROCESS macro and the specific models applied. It is necessary to provide a more detailed interpretation of the correlations found between the main variables, explaining how these relationships support or refute the hypotheses put forward. Data visualization can be improved by including additional graphs illustrating the key relationships and the mediation and moderation effects discussed in the study.

Response:

Thank you for your valuable feedback. We have made several revisions to address your comments: We have provided a detailed justification for the choice of the PROCESS macro and the specific models applied in our analysis. Additionally, we have expanded the interpretation of the correlations found between the main variables, explaining how these relationships support or refute the hypotheses put forward. Furthermore, we have included more graphs to illustrate the key relationships and the mediation and moderation effects discussed in the study, thereby improving data visualization. We appreciate your suggestions and believe these changes enhance the clarity and comprehensiveness of our manuscript.

Comments 4: In the conclusions section, it is important to expand on the practical and theoretical implications, providing specific recommendations for future interventions based on the study findings. Also expand on the discussion of the limitations of the study, detailing how the cultural context and sampling may have influenced the results. Proposing clear directions for future research, including suggestions for additional longitudinal studies and alternative methodological approaches, will contribute to a better understanding of the phenomenon under investigation.

Response:

Thank you for your insightful feedback. We have supplemented the conclusions section with specific recommendations for future interventions based on the study findings. Additionally, we have expanded the discussion of the study's limitations, detailing how the cultural context and sampling methods may have influenced the results. Furthermore, we have proposed clear directions for future research, including suggestions for additional longitudinal studies and alternative methodological approaches.

Comments 5: Comments on the Quality of English Language

It is crucial to ensure that the text is coherent and clear throughout, avoiding excessive use of jargon and explaining technical terms when necessary. Finally, it is recommended to verify that all citations and references are complete and correctly formatted according to the journal's style, thus ensuring the accuracy and professionalism of the article.

Response:Thank you for your valuable feedback on the quality of the English language in our manuscript. We have thoroughly reviewed the text to ensure coherence and clarity, minimizing the use of jargon and providing explanations for technical terms where necessary. Additionally, we have verified that all citations and references are complete and correctly formatted according to the journal's style, ensuring the accuracy and professionalism of our article. We appreciate your constructive suggestions and believe they have contributed to the improvement of our manuscript.

Reviewer 2 Report

Comments and Suggestions for Authors

The objective of this study is to examine the longitudinal associations, underlying mechanisms, and gender differences between the FoC-19 and IGA within the context of Chinese culture.  Research on the effects of COVID-19 is still ongoing and can bring new insights into the scientific field. Therefore I enjoyed while reading this manuscript. 

The rationale was well-described by the authors, and the study is methodologically sound in that it effectively analyses the long term COVID effects and associated variables it deals with.  

It would be more logical to present the research hypotheses collectively under the conceptual model on page 4.  Please write all of your hypotheses after the figure.

The methodology and analysis employed in the research are robust and well-explicated. However, the sample used in the study is problematic. The authors elected to utilise a convenience sample, which is the least robust and therefore the least suitable for generating generalisations.  The issue regarding the sample was mentioned in the limitation section. I recommend the authors to add a sentence about the limitiation of the sample selection method.

It is also crucial to ascertain the total number of students included in the research sample. Please provide the exact figure to ensure that the sample accurately reflects the population. 

I was pleased to see the details regarding the scales used in the research, as well as the Cronbach alpha values. The authors also declared that their data was normally distributed. However, I would like to enquire as to whether the authors conducted confirmatory factor analysis (CFA) before testing their model. If this was done, please provide the CFA results. 

I am pleased to report that I found the authors' presentation of the research findings to be highly effective. The results were conveyed in a clear and straightforward manner. However, the discussion section could be strengthened to better align with the other strengths of the research. It would be beneficial to contextualise the results within the broader research landscape on the effects of the Coronavirus. 

Finally, the implications section should be reinforced by providing concrete recommendations to policy makers. While the authors have successfully outlined implications for school administrators, it would be beneficial to also include recommendations for policy makers and governments.

Author Response

Thank you very much for taking the time to review this manuscript. Please find the detailed responses below and the corresponding revisions in the re-submitted files.

Comments 1: It would be more logical to present the research hypotheses collectively under the conceptual model on page 4.  Please write all of your hypotheses after the figure.

Response:Thank you for your constructive feedback. We have revised the manuscript to present all research hypotheses collectively under the conceptual model on page 4. All hypotheses are now clearly stated after the figure, ensuring a more logical flow and better organization of the content. We appreciate your suggestion and believe this change enhances the clarity and coherence of our manuscript.

Comments 2:The methodology and analysis employed in the research are robust and well-explicated. However, the sample used in the study is problematic. The authors elected to utilise a convenience sample, which is the least robust and therefore the least suitable for generating generalisations.  The issue regarding the sample was mentioned in the limitation section. I recommend the authors to add a sentence about the limitiation of the sample selection method. It is also crucial to ascertain the total number of students included in the research sample. Please provide the exact figure to ensure that the sample accurately reflects the population.

Response:

Thank you for your insightful feedback. We acknowledge the limitations associated with the use of a convenience sample. In response to your suggestion, we have added the following sentence in the limitations section to explicitly address the limitations of the sample selection method: "Due to the research design adopting a longitudinal approach, we focused on the university student population, considering the accessibility of participants, potential attrition, and research costs." We appreciate your recommendation and believe this addition provides a more comprehensive understanding of the study's limitations. Moreover, the total number of students included in the study is now clearly stated in the manuscript.

Comments 3: I was pleased to see the details regarding the scales used in the research, as well as the Cronbach alpha values. The authors also declared that their data was normally distributed. However, I would like to enquire as to whether the authors conducted confirmatory factor analysis (CFA) before testing their model. If this was done, please provide the CFA results.

Response:

Thank you for your positive feedback and your inquiry regarding the confirmatory factor analysis (CFA). We appreciate your attention to detail. We have now included the CFA results in the manuscript to provide a more comprehensive overview of our analysis. The results confirm the validity of the measurement model.

3.2 Testing the measurement model

This research extensively evaluated the measurement model's fit by employing well-established indices such as the chi-square test, comparative fit index (CFI), Tucker-Lewis index (TLI), root mean square error of approximation (RMSEA), and standardized root mean square residual (SRMR). These metrics are well-regarded in the field as indicators of model adequacy. The overall fit of the measurement model was assessed using Mplus 8.3 software. Table 3 demonstrates that the goodness-of-fit indices for all scales indicated acceptable construct validity. These results collectively confirm the satisfactory fit of the measurement model across different scales in our study.

Table 3 The goodness of fit of the measurement model

χ2/df

CFI

TLI

SRMR

RMSEA

1.FoC-19 scale

3.04

0.97

0.94

0.032

0.079

2.Loneliness scale

2.24

0.96

0.94

0.041

0.063

3.The PHQ-9

2.92

0.95

0.93

0.027

0.078

4.IGA scale

2.75

0.95

0.94

0.036

0.074

Comments 4:I am pleased to report that I found the authors' presentation of the research findings to be highly effective. The results were conveyed in a clear and straightforward manner. However, the discussion section could be strengthened to better align with the other strengths of the research. It would be beneficial to contextualise the results within the broader research landscape on the effects of the Coronavirus.

Response:These revisions enhance the discussion by situating our findings within the wider research context and highlighting the relevance and implications of our study in relation to the broader effects of the COVID-19 pandemic.

To contextualize our results within the broader research landscape, it is important to note that the psychological impacts of the COVID-19 pandemic have been widely documented, with numerous studies highlighting increased levels of anxiety, depression, and stress among various populations. Our study adds to this body of literature by specifically examining the role of FoC-19 in the development of IGA among university students. This aligns with previous research suggesting that the pandemic has exacerbated pre-existing mental health issues and introduced new stressors that contribute to maladaptive coping mechanisms, such as excessive online gaming.

Comments 5:.Finally, the implications section should be reinforced by providing concrete recommendations to policy makers. While the authors have successfully outlined implications for school administrators, it would be beneficial to also include recommendations for policy makers and governments.

Response:In addition to these recommendations for school administrators, policymakers and governments should consider implementing broader strategies to support the mental health of university students. This could include funding for mental health programs, developing policies that promote mental health awareness and education, and ensuring that universities have the resources necessary to provide comprehensive mental health services. Moreover, policies that foster a supportive online environment and regulate the gaming industry to prevent addictive game designs could help mitigate the risks of IGA. By addressing both the individual and systemic factors contributing to IGA, these measures can create a more supportive environment for the mental health and academic success of college students.

Reviewer 3 Report

Comments and Suggestions for Authors

The article presented is interesting for the scientific community. After the pandemic, an increase in cases of worsening psychological well-being has been recorded. The research serves to carry out a program to improve psychological well-being by universities.

The arguments and discussion of the findings are coherent and convincing.

The relationship between theory and methodology is appropriate because it applies the Theory of compensatory use of the Internet and the theory of social role.

There is one issue that the authors must specify: is their research should replace Gender with Sex. The authors have indicated that there are two options (male or female), this is typical of the Sex and not Gender option. The authors have to justify why they have put Gender and not Sex.

The authors have correctly interpreted the results of the analyzes. For the future, I recommend using SPSS to carry out the moderated multiple mediation model because the program has interesting options for statistical analysis.

I recommend citing the following articles as they have the same line of research and European comparison.

- Counselling services and mental health for international chinese college students in post-pandemic thailand. Education Sciences, 12(12), 866. https://doi.org/10.3390/educsci12120866

- Challenge-obstacle stressors and cyberloafing among higher vocational education students: the moderating role of smartphone addiction and Maladaptive. Frontiers in Psychology, 15, 1358634. https://doi.org/10.3389/fpsyg.2024.1358634

I recommend briefly expanding the Conclusions section

Author Response

Thank you very much for taking the time to review this manuscript. Please find the detailed responses below and the corresponding revisions in the re-submitted files.

Comments 1:  There is one issue that the authors must specify: is their research should replace Gender with Sex. The authors have indicated that there are two options (male or female), this is typical of the Sex and not Gender option. The authors have to justify why they have put Gender and not Sex.

Response:Thank you for your insightful feedback. We understand the importance of distinguishing between "gender" and "sex" in research. Here is our rationale for using "gender" instead of "sex" in our study:

Firstly, our choice to use "gender" aligns with the conventions of previous related studies in this field, which predominantly use "gender" to discuss differences in behaviors, perceptions, and psychological aspects. This helps maintain consistency and coherence with existing literature.

Secondly, while "sex" refers to biological differences between males and females, such as physical characteristics and genetic differences, "gender" emphasizes the psychological and social differences, including gender roles, identity, and stereotypes. Given that our study focuses on psychological constructs like FoC-19, loneliness, depression, and IGA, which are influenced by social and cultural factors, "gender" is a more appropriate term. These constructs are often shaped by societal expectations and norms related to gender, making "gender" a more relevant and comprehensive descriptor for our research context.

Comments 2: I recommend citing the following articles as they have the same line of research and European comparison.

Response:Thank you for your valuable suggestion. We have cited the recommended articles in the discussion section to further enrich the content of our paper. We appreciate your recommendation and believe these additions enhance the relevance and depth of our study.

82.Cao, H., Lin, W., Chen, P. Counselling services and mental health for international Chinese college students in post-pandemic thailand. Education Sciences, 2022, 12, 866. https://doi.org/10.3390/educsci12120866

83.Lizarte Simón EJ, Khaled Gijón M, Galván Malagón MC, Gijón Puerta J. Challenge-obstacle stressors and cyberloafing among higher vocational education students: the moderating role of smartphone addiction and Maladaptive. Front Psychol. 2024, 15, 1358634. https://doi.org/10.3389/fpsyg.2024.1358634

Comments 3: I recommend briefly expanding the Conclusions section

Response:Thank you for your suggestion. We have expanded and refined the content of the Conclusions section accordingly.

This study is the first to utilize a longitudinal design to demonstrate a positive association and the underlying mechanisms between FoC-19 and IGA over time. Our research extends the application of the CIUT and social role theory to the domain of addictive behaviors, providing new insights into how these theories explain the relationship between pandemic-related stressors and IGA. Our findings reveal that loneliness and depression serve as key mediators in the relationship between FoC-19 and IGA, highlighting the importance of addressing these emotional factors in interventions aimed at reducing IGA among young adults. Additionally, the moderating effect of gender indicates that the predictive effect of FoC-19 on IGA is significant only for male students, suggesting that gender-specific approaches may be necessary for effective intervention. In summary, our study not only contributes to the theoretical understanding of the links between FoC-19 and IGA but also provides practical implications for the design of interventions aimed at supporting young adults during and after the pandemic. Future research should continue to explore these relationships and examine the effectiveness of proposed intervention strategies in diverse populations and settings.

Round 2

Reviewer 3 Report

Comments and Suggestions for Authors

All modifications have been made by the authors. In my opinion, the article meets all the criteria for publication.